# COARFORMER: TRANSFORMER FOR LARGE GRAPH VIA GRAPH COARSENING

## ABSTRACT

Although Transformer has been generalized to graph data, its advantages are mostly observed on small graphs, such as molecular graphs. In this paper, we identify the obstacles of applying Transformer to large graphs: (1) The vast number of distant nodes distract the necessary attention of each target node from its local neighborhood; (2) The quadratic computational complexity regarding the number of nodes makes the learning procedure costly. We get rid of these obstacles by exploiting the complementary natures of GNN and Transformer, and trade the fine-grained long-range information for the efficiency of Transformer. In particular, we present Coarformer, a two-view architecture that captures fine-grained local information using a GNN-based module on the original graph and coarse yet long-range information using a Transformer-based module on the coarse graph (with far fewer nodes). Meanwhile, we design a scheme to enable message passing across these two views to enhance each other. Finally, we conduct extensive experiments on real-world datasets, where Coarformer outperforms any single-view method that solely applies a GNN or Transformer. Besides, the coarse global view and the cross-view propagation scheme enable Coarformer to perform better than the combinations of different GNN-based and Transformer-based modules while consuming the least running time and GPU memory.

## 1 INTRODUCTION

In recent years, the Transformer architecture (Vaswani et al., 2017) has been derived into several variants, e.g., BERT (Devlin et al., 2019) and ViT (Dosovitskiy et al., 2021), which achieve unprecedented successes in natural language processing (NLP) and computer vision (CV), respectively. Some recent works (Kreuzer et al., 2021; Ye & Ji, 2021; Ying et al., 2021a) attempt to generalize Transformer for graph data by treating each node as a token and designing dedicated positional encoding for the nodes. These works' performance has surpassed that of graph neural networks (GNN) on an increasing number of graph-related tasks, particularly molecular property prediction (Ying et al., 2021b).

These works, however, demonstrate their superiority on small graphs, such as molecules with tens of atoms (i.e., nodes). When applied to a large graph, these Transformer-based methods (Dwivedi & Bresson, 2021; Zhang et al., 2020) explicitly or implicitly restrict each node's receptive field to its neighbors. Nevertheless, large graphs, such as Arxiv and Products in OGBN (Hu et al., 2020), also demand the global receptive field and the powerful expressiveness of the Transformer architecture. These graphs have more than hundreds of thousands of nodes and graph diameters larger than twenty, where the needed receptive field should be large, and the correlations among the nodes' features are complex.

So, what limits the applicability of Transformer on large graphs? Here, we identify two main obstacles: (1) Transformer is built on a node-to-node attention mechanism. When applied to a large graph with numerous nodes, the massive distant nodes can divert a significant portion of the attention no matter whether they are indeed related. As a result, the target node can neglect its local neighborhood, which is indispensable for learning generalizable node representations. Hence, Transformer is prone to causing over-fitting on large graphs, which will be exacerbated for the semi-supervised node-level tasks where the number of labeled nodes is limited. We will provide some empirical evidences for this point in Sec. 5.1.1 and Appendix A.1. (2) The global receptive field of Transformer is

costly, where the pairwise interactions among tokens lead to quadratic computational complexity regarding the number of nodes. Although some Sparse Transformer methods (Roy et al., 2021; Kitaev et al., 2019; Ren et al., 2021) can improve the efficiency of the original Transformer, they have not exploited the unique characteristics of graph data and require a quadratic or at least sub-quadratic space complexity, which is still unaffordable in most practical cases.

We argue that both local and global information is useful for encoding each node in a large graph. Meanwhile, we notice that GNN and Transformer are proficient at capturing the local and global information, respectively, but neither of them can easily extract both kinds of information. Thus, a straightforward strategy is to employ them together to play to their strengths simultaneously. To this end, we propose to wipe out the discussed obstacles of applying Transformer by letting it work on a down-sampled graph that preserves both the sketch of global graph structure and the aggregated node features yet has much fewer nodes. And the similar idea is used in graph pooling (Ying et al., 2018; Baek et al., 2020) and hierarchical GNN (Fang et al., 2020; Sobolevsky, 2021). In this way, we essentially trade some fine-grained long-range information for the efficiency of Transformer.

To encode both local and global information for each node, we propose a two-view architecture Coarformer consisting of a fine-grained local view and a coarse global view. In the local view, we can apply a GNN-based module to the original input graph to encode each node by its local topological structures and the node features. In the global view, we apply a Transformer-based module to a coarse graph produced by an adopted graph coarsening algorithm. Such kinds of algorithms mimic a down-sampling to the original graph via grouping the nodes into a less number of super-nodes. The Transformer-based module makes pairwise interactions among these super-nodes, capturing the coarse but long-range dependencies of the original graph. Additionally, we design a cross-view propagation strategy for these two views in order to facilitate their interaction. As a result, Coarformer encodes each node by both its local and global information, combining GNN and Transformer's merits. It is worth noting that the number of super-nodes balances the efficiency of the Transformer-based module and how coarse the global view is. In practice, a small number that reduces the computational complexity from quadratic to linear w.r.t. the number of original nodes can still provide helpful global information, making Coarformer applicable for large graphs.

We conduct extensive experiments on real-world datasets to study the proposed Coarformer. At first, the combinations of a GNN model with various Transformer-based methods consistently surpass a single GNN model, which supports our idea of encoding nodes by local and global information. More importantly, Coarformer achieves the best performances with the least running time and GPU memory consumption, compared to the related Sparse Transformer-based methods. Meanwhile, we empirically show that the local and global information is complementary, and our coarse global view can boost the performances of various kinds of GNNs.

## 2 BACKGROUND AND MOTIVATIONS

In this section, we summarize GNN, Transformer for graphs, and graph coarsening, along with a discussion on their connections to motivate our method. Before that, we firstly present notations for graph data. Let $\mathcal{G} = (\mathcal{V}, \mathcal{E})$ be a undirected unweighted graph, where $\mathcal{V}$ is the node set, $\mathcal{E}$ is the edge set. We denote the number of nodes (i.e., $|\mathcal{V}|$) by $n$ and use $\boldsymbol{X} \in \mathbb{R}^{n \times k_0}$ to denote the feature matrix whose $i$-th row $\boldsymbol{X}_{i,:}$ is the feature vector of node $v_i$. We use $\boldsymbol{A} \in \{0,1\}^{n \times n}$ to denote the adjacency matrix of $\mathcal{G}$, where $\boldsymbol{A}_{ij} = 1$ if and only if $(v_i, v_j) \in \mathcal{E}$. The degree of a node $v_i$ is denoted by $d_i = \sum_{j=1}^{n} \boldsymbol{A}_{ij}$, and the degree matrix $\boldsymbol{D}$ is a diagonal matrix with $d_i$ being its $i$-th diagonal entry.

**GNN.** To represent nodes based on their features (i.e., $\boldsymbol{X}$) and topological structures (i.e., $\boldsymbol{A}$), most GNN models stack the message passing layers to calculate node representations $\boldsymbol{H}^{(l)}$ by recursively aggregating the representations of its neighbors. Taking GCN (Kipf & Welling, 2017) for example, it defines this procedure as $\boldsymbol{H}^{(0)} = \boldsymbol{X}, \boldsymbol{H}^{(l)} = \tilde{\boldsymbol{D}}^{-\frac{1}{2}} \tilde{\boldsymbol{A}} \tilde{\boldsymbol{D}}^{\frac{1}{2}} \boldsymbol{H}^{(l-1)} \boldsymbol{W}^{(l)}$, where $\tilde{\boldsymbol{D}} = \boldsymbol{D} + \boldsymbol{I}_n$, $\tilde{\boldsymbol{A}} = \boldsymbol{A} + \boldsymbol{I}_n$, and $\boldsymbol{W}^{(l)} \in \mathbb{R}^{k_{(l-1)} \times k_l}$ is a learnable parameter. In such a way, an $L$-layer GNN model calculates $\boldsymbol{H}^{(L)}$ to represent the nodes, where the receptive field for each node is its $L$-hop neighborhood. To capture long-range information, we need to increase $L$, where a large depth $L$ is prone to causing over-smoothing (Li et al., 2018) and bottleneck (Alon & Yahav, 2021) issues.

**Transformer for graphs.** Therefore, some recent works apply the Transformer architecture to capture useful long-range information in the graph, which relies solely on the attention mechanism,

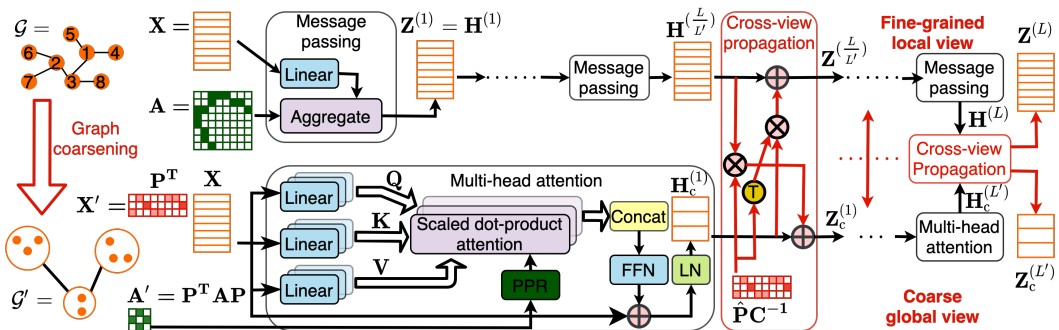

Figure 1: Overview of the Coarformer architecture.

dispensing with the message passing used in GNN. Taking the scaled dot-product attention for example, the node representations are updated as the weighted sums of themselves:

$$
\boldsymbol{H}^{(l)} = \mathrm{Softmax}\left(\frac{\boldsymbol{Q}\boldsymbol{K}^{\mathrm{T}}}{\sqrt{m}}\right)\boldsymbol{V} = \mathrm{Softmax}\left(\frac{(\boldsymbol{H}^{(l-1)}\boldsymbol{W}_Q^{(l)})(\boldsymbol{H}^{(l-1)}\boldsymbol{W}_K^{(l)})^{\mathrm{T}}}{\sqrt{m}}\right)\boldsymbol{H}^{(l-1)}\boldsymbol{W}^{(l)}, \quad (1)
$$

where $\boldsymbol{W}_Q, \boldsymbol{W}_K \in \mathbb{R}^{k_{(l-1)} \times m}$, and $\boldsymbol{W} \in \mathbb{R}^{k_{(l-1)} \times k_l}$ are learnable parameters. Comparing Eq. 1 with the updating equation used in GCN, $\mathrm{Softmax}\left(\frac{\boldsymbol{Q}\boldsymbol{K}^{\mathrm{T}}}{\sqrt{m}}\right)$ is usually allowed to be a dense $n \times n$ matrix, while the $(i, j)$-th entry of $\tilde{\boldsymbol{D}}^{-\frac{1}{2}}\tilde{\boldsymbol{A}}\tilde{\boldsymbol{D}}^{\frac{1}{2}}$ is non-zero only if $\tilde{\boldsymbol{A}}_{ij} \neq 0$. Thus, one attention layer is sufficient for each $\boldsymbol{h}_j^{(l-1)}$ to attend to any $\boldsymbol{h}_i^{(l)}$, no matter how far away $v_j$ is to $v_i$. Although these pairwise interactions provide the global receptive field for Transformer to capture long-range information, they often distract the necessary attention of each node from its neighborhood, especially when $n$ is large. Besides, these pairwise interactions require $O(n^2)$ computational complexity, which further hinders the application of Transformer to large graphs.

**Graph coarsening.** To simplify a given graph (mainly reducing the number of nodes) while preserving its global information as much as possible, graph coarsening (Ron et al., 2011; Loukas, 2019) produces a partition $\mathcal{P} = \{\mathcal{C}_1, \ldots, \mathcal{C}_{n'}\}$ of $\mathcal{V}$ and regards each cluster $\mathcal{C}_i$ as a super-node. Consequently, we get a coarse graph $\mathcal{G}' = (\mathcal{V}', \mathcal{E}')$, where $|\mathcal{V}'| = n'$ and $(\mathcal{C}_i, \mathcal{C}_j) \in \mathcal{E}'$ if and only if $\exists v_k \in \mathcal{C}_i, v_l \in \mathcal{C}_j$, s.t. $(v_k, v_l) \in \mathcal{E}$. Meanwhile, this partition can be characterized by a matrix $\hat{\boldsymbol{P}} \in \{0, 1\}^{n \times n'}$, with $\hat{\boldsymbol{P}}_{ij} = 1$ if and only if $v_i \in \mathcal{C}_j$. Then its normalized version can be defined by $\boldsymbol{P} = \hat{\boldsymbol{P}}\boldsymbol{C}^{-\frac{1}{2}}$, where $\boldsymbol{C}$ is a $n' \times n'$ diagonal matrix with $|\mathcal{C}_i|$ as its $i$-th diagonal entry. We can define the feature matrix and weighted adjacency matrix for $\mathcal{G}'$ by $\boldsymbol{X}' = \boldsymbol{P}^{\mathrm{T}}\boldsymbol{X}$ and $\boldsymbol{A}' = \boldsymbol{P}^{\mathrm{T}}\boldsymbol{A}\boldsymbol{P}$, which characterize a coarse global view of the original graph. We define the coarsening rate as $c = \frac{n'}{n}$.

## 3 METHODOLOGY

In this section, we introduce a two-view architecture Coarformer composed of a fine-grained local view and a coarse global view. We present the overview of Coarformer in Figure 1. At first, we preprocess an input graph $\mathcal{G}$ via graph coarsening to generate a coarse graph $\mathcal{G}'$. Then a GNN-based module and a Transformer-based module work on $\mathcal{G}$ and $\mathcal{G}'$, respectively, which can be interpreted as encoding the nodes from the fine-grained local view and the coarse yet global view. We further design a scheme for Coarformer to enable neural messages to propagate across these two views, so the resulting node representations synthesize both the local and the global information.

In the fine-grained local view, any GNN model (e.g., GCN) can be adopted, which recursively calculates the node representations $\boldsymbol{H}^{(l)}, l = 1, \ldots, L$. As for the coarse global view, the details about the Transformer-based module are presented in Sec. 3.1. Then we elaborate on how to enhance the communication between these two views in Sec. 3.2. Lastly, we present how Coarformer makes inference and how to optimize Coarformer in Sec. 3.3.

### 3.1 COARSE GLOBAL VIEW

We apply a Transformer-based module to the coarse graph $\mathcal{G}'$, where each super-node is regarded as a token. Thus, we use $\boldsymbol{X}'$ as the initial token representation $\boldsymbol{H}_c^{(0)}$, where the subscript "c" stands for "coarse" to distinguish a matrix from its counterpart in our fine-grained local view. Then $\boldsymbol{H}_c^{(0)}$ is fed into the Transformer-based module, which consists of $L'$ layers of multi-head attention and produces $\boldsymbol{H}_c^{(l)}, l = 1, \ldots, L'$. In each multi-head attention layer, we use $h$ heads, i.e., $h$ scaled dot-product attention, each of which has its specific parameters and processes its input as Eq. 1: $\text{head}_i^{(l)} = \text{Softmax}\left(\frac{(\boldsymbol{H}_c^{(l-1)}\boldsymbol{W}_Q^{(l,i)})(\boldsymbol{H}_c^{(l-1)}\boldsymbol{W}_K^{(l,i)})^{\mathrm{T}}}{\sqrt{m}}\right)\boldsymbol{H}_c^{(l-1)}\boldsymbol{W}^{(l,i)}, i = 1, \ldots, h$. The outputs of the heads are concatenated and fed into a fully-connected feed-forward network (FFN) with ReLU activation in between its linear layers. We set the FFN of each layer to ensure the token representation at each layer has the same dimension, i.e., $k_1 = \cdots = k_{L'}$, which allows us to consider a residual connection between each consecutive layer. Finally, a layer normalization (LN) is applied, and then the output of the $l$-th multi-head attention layer is: $\boldsymbol{H}_c^{(l)} = \text{LN}(\text{FFN}(\text{Concat}(\text{head}_1^{(l)}, \ldots, \text{head}_h^{(l)})) + \boldsymbol{H}_c^{(l-1)})$.

Up to now, $\boldsymbol{H}_c^{(l)}, l = 1, \ldots, L'$ depend only on the node features $\boldsymbol{X}'$. To utilize the topological structures of $\mathcal{G}'$, we employ personalized page rank (PPR) (Page et al., 1999) as a bias term for our attention weight matrix. Specifically, we first calculate the PPR matrix $\boldsymbol{R}' \in \mathbb{R}^{n' \times n'}$ based on the adjacency matrix $\boldsymbol{A}'$, where $\boldsymbol{R}'_{i,j}$ denotes the probability of starting a random walk with super-node $\mathcal{C}_i$ and stopping at $\mathcal{C}_j$. Then for each head at each layer, we calculate the attention weight matrix based on not only the inner products between queries and keys but also the PPR matrix: $\text{head}_i^{(l)} = \text{Softmax}\left(\frac{(\boldsymbol{H}_c^{(l-1)}\boldsymbol{W}_Q^{(l,i)})(\boldsymbol{H}_c^{(l-1)}\boldsymbol{W}_K^{(l,i)})^{\mathrm{T}}}{\sqrt{m}} + \beta_i^{(l)}\boldsymbol{R}'\right)\boldsymbol{H}_c^{(l-1)}\boldsymbol{W}^{(l,i)}$, where $\beta_i^{(l)}$ is a learnable parameter for adjusting the contribution of the PPR matrix.

**Computational complexity**. It is worth noting that the graph coarsening algorithms allow to control the size of coarse graph (i.e., $n'$). In essence, $n'$ balances the efficiency of the Transformer-based module and how coarse the global view is. We will empirically show in Sec. 5 that the useful global information can still be extracted from the coarse global view, even though $n'$ is as small as $\sqrt{n}$. Accordingly, the time complexity of our Transformer-based module is quadratic w.r.t. the number of super-nodes and thus linear w.r.t. the original number of nodes, i.e., $O(n'^2) = O(\sqrt{n}^2) = O(n)$. Meanwhile, the space complexity of our Transformer-based module is also $O(n)$, as the dominant attention weight matrix has size $n' \times n'$, which can fit into the GPU for most practical large graphs.

### 3.2 CROSS-VIEW PROPAGATION

As a two-view architecture, the node representations $\boldsymbol{H}^{(l)}, l = 1, \ldots, L$, calculated by the GNN-based module, encode the fine-grained local information of the nodes, while the representations of the super-nodes $\boldsymbol{H}_c^{(l)}, l = 1, \ldots, L'$ capture long-range dependencies among the nodes but with a coarse granularity. We argue that, to predict for the nodes, both their local and global information are helpful, and thus we should synthesize these representations encoded from the two views.

To this end, we design a scheme of cross-view propagation for Coarformer, where the encoded representations are allowed to propagate across these two views. At first, due to its pairwise interaction nature, the Transformer-based module does not need to be deep, thus it is safe to assume $L' \leq L$. Then we denote the outputs of each layer in these two modules by $\boldsymbol{Z}^{(l)}$ and $\boldsymbol{Z}_c^{(l')}$, respectively, and define their forward propagation as follow:

$$\boldsymbol{Z}^{(l)} = \begin{cases} \boldsymbol{H}^{(l)} + \hat{\boldsymbol{P}}\boldsymbol{C}^{-1}\boldsymbol{H}_c^{(l/\lfloor\frac{L}{L'}\rfloor)} & \text{if } l \bmod \lfloor\frac{L}{L'}\rfloor = 0, \\ \boldsymbol{H}^{(l)} & \text{elsewise;} \end{cases}$$

$$\boldsymbol{Z}_c^{(l')} = \boldsymbol{H}_c^{(l')} + (\hat{\boldsymbol{P}}\boldsymbol{C}^{-1})^{\mathrm{T}}\boldsymbol{H}^{(l'\lfloor\frac{L}{L'}\rfloor)}.$$

$$(2)$$

For simplicity, we do not explicitly re-define $\boldsymbol{H}^{(l+1)}$ and $\boldsymbol{H}_c^{(l'+1)}$ based on $\boldsymbol{Z}^{(l)}$ and $\boldsymbol{Z}_c^{(l')}$, respectively, but please keep in mind that $\boldsymbol{Z}^{(l)}$ and $\boldsymbol{Z}_c^{(l')}$ do serve as the inputs to their corresponding next layers. With this cross-view propagation, the outputs of our two modules depend on each other.

Thus, once we have properly optimized them, their outputs would properly synthesize both local and global information from the two views, leading to more discriminative node representations.

**Sampling-based cross-view propagation on large graphs**. Generally, full-batch training is infeasible on large graphs. With mini-batch training, only the rows of $\boldsymbol{H}^{(l'\lfloor\frac{L}{L'}\rfloor)}$ that correspond to the target nodes sampled at that layer are available, and thus we cannot calculate $\boldsymbol{Z}_c^{(l')}$ exactly. Luckily, a more careful look at Eq. 2 shows that, $\forall i = 1, \ldots, n'$, the $i$-th row of $\boldsymbol{Z}_c^{(l')}$ can be written as:

$$(\boldsymbol{Z}_c^{(l')})_{i,:} = (\boldsymbol{H}_c^{(l')})_{i,:} + \frac{1}{|\mathcal{C}_i|} \sum_{v_j \in \mathcal{C}_i} \boldsymbol{H}_{j,:}^{(l'\lfloor\frac{L}{L'}\rfloor)} = (\boldsymbol{H}_c^{(l')})_{i,:} + \mathbb{E}_{v_j \sim \text{Uniform}(\mathcal{C}_i)}[\boldsymbol{H}_{j,:}^{(l'\lfloor\frac{L}{L'}\rfloor)}]. \quad (3)$$

Suppose the nodes in each mini-batch are sampled by a widely adopted GraphSAINT sampler (Zeng et al., 2020) that is random walk-based with a "teleporting" probability $\alpha$. Let's denote the nodes sampled at $l$-th layer by $\mathcal{B}^{(l)}$. We can pre-compute the probability distributions for $v_i \in \mathcal{V}$: $P^{(l)}(v_i) = \frac{1}{n}(\alpha \boldsymbol{I}_n + (1-\alpha)\boldsymbol{D}^{-1}\boldsymbol{A})^{L-l}\boldsymbol{1}_n)_i, l = 1, \ldots, L$, and approximate $\boldsymbol{Z}_c^{(l')}$ as follow:

**Proposition 1** *Denoting* $\mathcal{B}_i^{(l)} = \{v | v \in \mathcal{C}_i \cap \mathcal{B}^{(l)}\}$, $\forall i \in \{1, \ldots, n'\}$, $l' = \{1, \ldots, L'\}$, $(\boldsymbol{H}_c^{(l')})_{i,:} +$

$\sum_{v_j \in \mathcal{B}_i^{(l'\lfloor\frac{L}{L'}\rfloor)}} \frac{1}{|\mathcal{C}_i|} \dfrac{\sum_{v_k \in \mathcal{B}_i^{(l'\lfloor\frac{L}{L'}\rfloor)}} P^{(l'\lfloor\frac{L}{L'}\rfloor)}(v_k)}{P^{(l'\lfloor\frac{L}{L'}\rfloor)}(v_j)} \boldsymbol{H}_{j,:}^{(l'\lfloor\frac{L}{L'}\rfloor)}$ *is an unbiased estimator for* $(\boldsymbol{Z}_c^{(l')})_{i,:}$.

The proof is simple: $P^{(l)}(v_i)$ is the sampling probability of $v_i$ at the $l$-th layer, and $\frac{1}{|\mathcal{C}_i|} \dfrac{\sum_{v_k \in \mathcal{B}_i^{(l'\lfloor\frac{L}{L'}\rfloor)}} P^{(l'\lfloor\frac{L}{L'}\rfloor)}(v_k)}{P^{(l'\lfloor\frac{L}{L'}\rfloor)}(v_j)}$ serves as the likelihood ratio used in importance sampling. Thus, Coarformer can cooperate with mini-batch training and thus is viable on large graphs.

### 3.3 INFERENCE AND OPTIMIZATION

Without loss of generality, we consider using Coarformer for node classification task. We denote node labels by $\boldsymbol{Y} \in \{0,1\}^{n \times |\mathcal{Y}|}$ where $\mathcal{Y}$ is the label set, $\boldsymbol{Y}_{i,:}$ is a one-hot row vector, and $\boldsymbol{Y}_{i,j} = 1$ indicates $v_i$ belongs to the $j$-th class. We can naturally deduce the labels of the super-nodes by $\boldsymbol{Y}' = \boldsymbol{P}^T\boldsymbol{Y}$ for our coarse graph $\mathcal{G}'$. To predict $\boldsymbol{Y}$ and $\boldsymbol{Y}'$, we can simply stack a linear output layer upon the final node representations. Specifically, the predictions made from our two views are $\hat{\boldsymbol{Y}} = \text{Softmax}(\boldsymbol{Z}^{(L)}\boldsymbol{W} + \boldsymbol{1}_n\boldsymbol{b}^T)$ and $\hat{\boldsymbol{Y}}' = \text{Softmax}(\boldsymbol{Z}_c^{(L')}\boldsymbol{W}' + \boldsymbol{1}_{n'}\boldsymbol{b}'^T)$, respectively, where $\boldsymbol{W} \in \mathbb{R}^{k_L \times |\mathcal{Y}|}$, $\boldsymbol{W}' \in \mathbb{R}^{k_{L'} \times |\mathcal{Y}|}$, $\boldsymbol{b} \in \mathbb{R}^{|\mathcal{Y}|}$, and $\boldsymbol{b}' \in \mathbb{R}^{|\mathcal{Y}|}$ are learnable parameters.

Denoting the loss function, e.g., Cross-Entropy loss, by $\mathcal{L}(\cdot, \cdot)$, we can learn the parameters of Coarformer by minimizing $\gamma\mathcal{L}(\hat{\boldsymbol{Y}}, \boldsymbol{Y}) + (1-\gamma)\mathcal{L}(\hat{\boldsymbol{Y}}', \boldsymbol{Y}')$ with $\gamma \in [0, 1]$ as a hyper-parameter for balancing the impacts of our two views. As our cross-view propagation scheme has enabled $\boldsymbol{Z}^{(L)}$ to contain both the local and the global information, we adopt $\gamma = 1$ in our experiment for simplicity, where the Transformer-based module is still optimized to cooperate with the GNN-based module.

## 4 RELATED WORK

**Transformer for graphs**. Recently, models based on the Transformer architecture have shown their superiority in more and more domains, e.g., BERT (Devlin et al., 2019) in NLP and ViT (Dosovitskiy et al., 2021) in CV. Existing works that attempt to generalize the Transformer architecture for graph data mainly focus on two problems: (1) How to design the positional encoding for nodes? (2) How to make the pairwise interactions computationally tractable on large graphs? For the positional encoding, Laplacian eigenvectors (Dwivedi & Bresson, 2021; Kreuzer et al., 2021), hop-based relative distance (Zhang et al., 2020; Ying et al., 2021a), and Weisfeiler-Lehman code (Zhang et al., 2020) have been studied from both theoretical and empirical aspects. For the scalability issue, an immediate workaround is to restrict the receptive field of each node, e.g., GAT (Veličković et al., 2018) and GT-sparse (Dwivedi & Bresson, 2021) consider just the 1-hop neighbors, which yet sacrifices one of the main advantages of Transformer. To make a better trade-off, ADSF (Zhang et al., 2019) relaxes this restriction to include high-order neighbors based on random walks. SGAT (Ye &

Ji, 2021) and SAC (Li et al., 2020) learn to select a subset of each node's high-order neighborhood. Graph-Bert (Zhang et al., 2020) restricts the receptive field of each node to the nodes with top-$k$ intimacy scores (e.g., Katz and PPR). However, all these restrictions possess a strong bias towards a node's local neighbors. Instead, Coarformer conducts the regular pairwise interactions for super-nodes, where long-range information can be easily captured and lifted back to the original nodes via our designed cross-view propagation.

**Sparse Transformer for NLP**. In parallel, many sparse Transformer methods have been proposed to reduce the computational complexity of the Transformer in the field of NLP. Longformer (Beltagy et al., 2020) applies block-wise or strode patterns, just like the GAT only focuses on specific neighbors. Reformer (Kitaev et al., 2019) introduces a hash-based similarity measure to cluster tokens into chunks efficiently. Routing Transformer (Roy et al., 2021) employs online k-means clustering on the tokens. Combiner (Ren et al., 2021) introduces a bi-level factorization of the attention weights, unifying the expression of most of the existing sparse transformers. However, these factorization-based methods often require a quadratic or sub-quadratic space complexity regarding the number of nodes, still making them infeasible on large graphs. By running a Transformer-based module on a coarse graph with $n' \approx \sqrt{n}$ nodes, Coarformer decreases this space complexity and thus saves a lot of GPU memory, as Figure 2 demonstrates.

## 5 EXPERIMENTS

In this section, we conduct extensive experiments on real-world datasets to evaluate the performance of Coarformer. All the experiments are conducted on a machine with an NVIDIA RTX 2080ti GPU (11GB), Intel Xeon Platinum 8163 CPU (2.50GHz), and 400GB of RAM.

### 5.1 PERFORMANCE COMPARISON

**Datasets**. Following GPR-GNN (Chien et al., 2020), we conduct our experiments on ten real-world datasets, including five homophilic graphs Cora, CiteSeer, PubMed (Sen et al., 2008; Yang et al., 2016), Computers, Photo (McAuley et al., 2015; Shchur et al., 2018), and five heterophilic graphs, Chameleon, Squirrel, Actor, Texas, Cornell (Rozemberczki et al., 2021; Tang et al., 2009; Pei et al., 2019). A detailed description of these datasets is provided in Appendix A.2.

**Protocol**. To conduct the experiments uniformly and fairly, we split the nodes into train/valid/test sets, where the ratio is 60%:20%:20%, and we randomly generate ten such splits for each dataset. Each method is evaluated with these ten splits for each dataset, and we report the averaged metric. To compare the performances and resource requirements of the considered methods, we choose accuracy, training time, and consumed GPU memory as the metrics. We perform hyperparameter optimization (HPO) for all methods, where the details can be found in Appendix A.3.1.

#### 5.1.1 OVERALL RESULTS

First, we aim to verify the benefits of combining a Transformer-based module working in the global view and a GNN-based module working in the local view. Second, we compare Coarformer with several related Transformer-based methods working in the global view without coarsening under our two-view architecture.

**Baselines**. For simplicity and generality, we consider GraphSAGE (Hamilton et al., 2017) working in the local view to accommodate full batch training and mini-batch training with which Transformer-based modules cooperate in the global view. Meanwhile, we use Graph Transformer (GT) as local model, which attends only to its neighbors, as an alternative model in the fine-grained local view for comparison. And for global model, we adapt the Transformer-based module Graphormer, which achieves excellent results in OGB-LSC (Hu et al., 2021), with different positional encoding and attentional bias: None, PPR, distance of the shortest path (SPD), PPR+SPD. Besides, under our two-view architecture, we select several related Transformer-based modules in the global view without coarsening combined with GraphSAGE in the local view as baselines to compare their performance with Coarformer and Coarformer without cross-view propagation called Coarformer (-CP): Graphormer with quadratic complexity; Graph-Bert, which uses top-$k$ PPR of each node as candidate set; Reformer, which uses locality sensitive hash (LSH) to reduce compu-

tational complexity; and Routing Transformer, which uses online $k$-means clustering to reduce the candidate set.

**Results and analysis**. We present the results about performance comparisons on Cora, CiteSeer, Pubmed, Chameleon, Actor, and Squirrel in Table 1. Results on the other datasets are presented in Appendix A.5.1. (1) Overall, the methods with two views (i.e., "Local+Global") outperform both the local view alone and the global view alone methods on almost all the datasets. This phenomenon suggests that the Transformer-based modules are skilled in capturing long-range information, which complements what the local view focuses on. (2) Among these "Local+Global" methods, Coarformer achieves the best performance on most of the datasets, demonstrating its effectiveness. Interestingly, although its Transformer-based module works on the original graph that provides more fine-grained global information than a coarse graph, +Graphormer (all) leads only on one dataset, less than Coarformer. Taking this comparison and the relatively poor performances of global view alone methods into consideration, we attribute the advantage of Coarformer to the coarse graph, which might implicitly regularize the Transformer-based module. Meanwhile, Coarformer outperforms the +Sparse Transformer methods. On the one hand, +Graph-Bert determines the candidate set for each target node by Top-$k$ PPR in the global view, which restricts the receptive field of Transformer-based module. On the other hand, +Reformer and +Routing do not incorporate the graph structure, making their captured information overly dependent on node features. Consequently, their extracted information might be less complementary to the local view. (3) The performance of Coarformer (-CP) is slightly weaker than that of Coarformer since the lack of information exchange between the local view and the global view, which confirms the necessity of our proposed cross-view propagation scheme.

We present the results concerning efficiency in Table 2, where the total training time excludes the time for data pre-processing (e.g., on Cora 4s, on Chameleon 5s) and inference. Time consumed by mini-batch training is significantly increased by more than 100 times, which is often intolerable in practice. As a result, when Transformer adapts to large graphs, it either exceeds the memory limit with full batch training or consumes too much time with mini-batch training. It is worth emphasizing that Coarformer significantly reduces the training time concerning the quadratic Transformer, such as Graphormer. In contrast to +Graphormer, the time consumption increases linearly with the number of nodes (given a coarsening rate of $\sqrt{n}$), which does not change the time complexity of the GNN-based module by order of magnitude, as shown in Figure 2.

Table 1: Performance comparisons between different Transformer-based modules: Mean accuracy (%) ± 95% confidence interval. Boldface letters are used to mark the best results. Results on more datasets are summarized at Table 12 in Appendix A.5.1.

| | | Cora | CiteSeer | PubMed | Chameleon | Actor | Squirrel |
|---|---|---|---|---|---|---|---|
| Local | GraphSAGE | 87.42±0.78 | 78.47±1.08 | 88.88±0.30 | 64.40±1.39 | 38.79±0.67 | 43.94±1.30 |
| | GT | 86.42±0.82 | 78.80±0.5 | 88.75±0.16 | 57.86±1.20 | 40.23±0.69 | 52.89±0.51 |
| Global | Graphormer (w/o) | 65.29±2.33 | 77.54±0.27 | 85.35±0.44 | 37.68±1.09 | 37.88±0.83 | 25.82±0.44 |
| | Graphormer (PPR) | 66.70±1.12 | 76.59±0.48 | 85.95±0.31 | 37.29±2.19 | 40.20±1.38 | 25.61±0.46 |
| | Graphormer (SPD) | 71.76±5.26 | 76.62±0.54 | 88.01±1.29 | 36.98±2.34 | 38.71±1.36 | 25.57±0.61 |
| | Graphormer (All) | 77.41±4.30 | 76.18±1.16 | 88.24±1.50 | 36.81±1.96 | 38.21±0.52 | 25.13±0.33 |
| Local (GraphSAGE) + Global (Xformer) | +Graphormer (All) | 88.03±0.98 | **81.23±0.55** | 89.65±0.37 | 64.20±0.57 | 40.05±0.62 | 42.10±0.73 |
| | +Graph-Bert | 88.10±0.29 | 80.46±0.51 | 88.70±0.12 | **66.54±1.09** | 39.66±0.52 | 48.38±0.90 |
| | +Reformer | 87.21±0.63 | 78.57±0.90 | 88.73±0.33 | 64.16±1.89 | 39.39±1.01 | 43.36±1.04 |
| | +RT | 80.31±0.34 | 70.63±0.38 | 76.57±0.26 | 63.83±0.60 | 40.13±0.39 | 42.44±0.64 |
| | Coarformer (-CP) | 87.00±0.75 | 77.52±0.44 | 88.91±0.21 | 60.63±1.46 | 38.95±0.87 | 40.34±0.83 |
| | Coarformer | **88.69±0.82** | 79.20±0.89 | **89.75±0.31** | 66.48±1.46 | **40.70±1.12** | **54.27±1.15** |

Table 2: Efficiency comparisons between different Transformer-based modules: average training time per epoch (ms)/total training time (s). Underlined letters imply results with mini-batch training.

| | Cora | CiteSeer | PubMed | Chameleon | Actor | Squirrel |
|---|---|---|---|---|---|---|
| GraphSAGE | 3.01/0.94 | 4.09/1.09 | 4.88/2.97 | 7.19/1.67 | 4.45/0.90 | 32.85/6.64 |
| +Graphormer(All) | 10.09/2.11 | 15.22/3.17 | 23319.70/6995.91 | 14.34/2.91 | 37.27/7.54 | 48.41/9.81 |
| +Reformer | 13.71/3.84 | 15.57/4.05 | 37.05/18.46 | 21.53/4.85 | 23.72/4.79 | 46.82/9.51 |
| +RT | 572.48/226.31 | 449.96/186.34 | 2964.85/950.29 | 349.41/121.71 | 860.45/301.07 | 720.47/256.76 |
| Coarformer | 7.61/2.28 | 9.14/2.13 | 8.81/5.21 | 10.95/2.52 | 8.14/1.64 | 34.00/6.88 |

### 5.1.2 EFFECTIVENESS AND ROBUSTNESS OF COARFORMER'S GLOBAL VIEW

In the previous section, we have shown the superiority of Coarformer, where GraphSAGE is adopted as the module working in Coarformer's local view. To verify the generality of the benefits brought

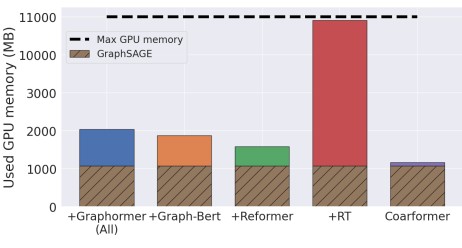 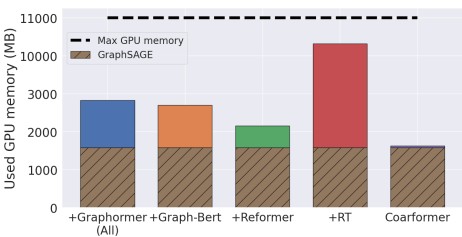

(a) on Cora          (b) on Chameleon

Figure 2: Comparisons about GPU memory usage.

in by the global view of Coarformer, we employ different GNNs and compare them to Coarformer with each of them as the module working in the local view of Coarformer. For this purpose, GCN, GIN (Xu et al., 2018), GAT, Graph Transformer (GT), APPNP (Klicpera et al., 2018), and GPR-GNN are considered.

**Results and Analysis**. We present the results about effectiveness in Table 3 on Cora, CiteSeer, PubMed, Chameleon, Actor, and Squirrel, where all the instances of Coarformer outperform their corresponding GNN models on most datasets. Results on other datasets are presented in Appendix A.5.2. Particularly, GIN performs much worse than other GNN models on several datasets, but with the global view of Coarformer, the gap is significantly reduced. Furthermore, Coarformer achieves the best performance on Cora, PubMed, Computers, Photo, Chameleon, Actor, and Squirrel. All these results confirm that the global view of Coarformer can boost the performance of an arbitrary GNN model, which further justifies the complementary natures of Coarformer's two views. Similarly, we conduct experiments on efficiency, where the detailed experimental results can be found in Appendix A.4. Whatever GNN-based module is in the local view, the time consumed by Coarformer remains in the same order of magnitude, where the increase is negligible.

Table 3: Performance comparisons on different GNN-based module: Mean accuracy (%) ± 95% confidence interval. Boldface letters are used to mark the improvements. Results on more datasets are summarized at Table 13 in Appendix A.5.2.

|          |      | GCN | GIN | GAT | GT | APPNP | GPR-GNN |
|----------|------|-----|-----|-----|-----|-------|---------|
| Cora     | w/o  | 87.06±0.63 | 84.11±0.82 | 87.18±0.66 | 86.42±0.82 | 88.10±0.73 | 88.48±0.51 |
|          | w/   | 87.64±0.65 | 85.63±0.83 | 87.83±0.76 | 86.91±0.62 | 88.78±0.62 | 88.93±0.31 |
|          | △    | **0.58** | **1.52** | **0.65** | **0.49** | **0.68** | **0.45** |
| CiteSeer | w/o  | 79.28±0.61 | 74.92±1.34 | 79.60±0.80 | 78.80±0.50 | 79.58±0.70 | 78.49±1.15 |
|          | w/   | 78.91±1.02 | 78.96±0.74 | 78.34±0.59 | 78.72±0.94 | 79.44±0.80 | 78.24±1.00 |
|          | △    | -0.27 | **4.04** | -1.26 | -0.08 | -0.14 | -0.25 |
| PubMed   | w/o  | 86.86±0.28 | 88.57±0.44 | 86.12±0.29 | 88.75±0.16 | 88.35±0.23 | 90.90±0.65 |
|          | w/   | 89.76±0.41 | 88.57±0.40 | 87.96±0.52 | 90.08±0.21 | 88.72±0.46 | 91.26±0.48 |
|          | △    | **2.90** | 0.00 | **1.84** | **1.33** | **0.37** | **0.36** |
| Chameleon | w/o | 58.80±0.90 | 38.84±2.55 | 59.41±1.55 | 57.86±1.2 | 53.76±1.44 | 66.63±1.41 |
|          | w/   | 67.64±1.29 | 63.85±2.36 | 67.59±1.88 | 66.59±0.86 | 59.85±0.99 | 67.09±1.53 |
|          | △    | **8.84** | **25.01** | **8.18** | **8.73** | **6.09** | **0.46** |
| Actor    | w/o  | 33.61±0.54 | 34.07±0.46 | 35.79±0.64 | 40.23±0.69 | 39.55±1.01 | 40.74±0.53 |
|          | w/   | 37.18±0.74 | 34.12±0.42 | 35.85±0.67 | 41.37±0.60 | 40.21±0.73 | 41.67±0.70 |
|          | △    | **3.57** | **0.05** | **0.06** | **1.14** | **0.66** | **0.93** |
| Squirrel | w/o  | 46.46±0.92 | 19.32±0.56 | 48.2±1.85 | 52.89±0.51 | 36.4±1.50 | 52.31±1.09 |
|          | w/   | 54.75±1.12 | 43.13±0.69 | 56.23±1.22 | 53.18±0.40 | 46.80±0.91 | 51.85±1.18 |
|          | △    | **8.29** | **23.81** | **8.03** | **0.29** | **10.40** | -0.46 |

## 5.2 COMPARISONS ON OGB

This experiment aims to verify the effectiveness of Coarformer on extremely large graphs.

**Datasets**. We adopt ogbn-arxiv and ogbn-products, wherein there are 169,343 and 2,449,029 nodes, respectively. We use the official train/valid/test splits for each dataset. Conventionally, we convert the directed graph ogbn-arxiv into an undirected graph. Details can be found in Hu et al. (2020).

**Protocol**. We largely follow the experimental setup in Sec. 5.1.1, yet all models are learned with mini-batch training due to the scale of the adopted datasets. Specifically, we employ GCN to cap-

ture the local information in the fine-grained local view, where random walk-based GraphSAINT is used to sample subgraphs for training the GCN. Then, we regard GCN as our testbed for comparing Coarformer with several Transformer-based methods under our two-view architecture, including Reformer and Routing Transformer (RT). Graphormer is excluded because calculating the PPR matrix and the SPD matrix for such huge graphs is intolerable. In addition, we perform HPO for all these methods. Please see Appendix A.3.2 for details.

**Results and Analysis** We present the results about effectiveness in Table 4, where Coarformer surpasses all the methods significantly (p-values = 0.02). The experimental results show that Coarformer can also improve performance of capturing long-range information significantly compared to GCN on extremely large graphs, wherein long-range information in the global view contributes. Moreover, we have to make the candidate set smaller because of the GPU memory limitation so that +Reformer and +RT even underperform GCN. These results remain consistent with the section 5.1.1.

Table 4: Performance comparison on OGB: Mean accuracy (%) ± 95% confidence interval. Boldface letters are used to mark the best results.

|  | GCN | +Reformer | +RT | Coarformer |
|---|---|---|---|---|
| ogbn-arxiv | 71.32±0.25 | 69.62±0.24 | 67.07±0.12 | **71.66±0.24** |
| ogbn-products | 78.70±0.34 | 74.09±0.23 | OOM | **79.18±0.20** |

## 5.3 SENSITIVITY ANALYSIS

Since Coarformer applies the Transformer-based module on a coarse graph, a question naturally comes up—Are the performances of Coarformer sensitive to the choice of coarsening algorithm as well as the adopted coarsening rate? Therefore, we evaluate Coarformer with different coarsening algorithms and different coarsening rates. Considered coarsening algorithms include Variation Neighborhoods (VN) (Loukas, 2019), Variation Edges (VE) (Loukas, 2019), and Algebraic JC (Alg JC) (Ron et al., 2011). Experimental settings are deferred to Appendix A.3.3.

**Results and Analysis**. We present the results in Table 5, where there is no apparent difference between different coarsening algorithms, indicating the robustness of Coarformer w.r.t. them. Overall, VN usually leads to higher accuracy on the test set compared to other coarsening algorithms, but other coarsening algorithms are also acceptable. Interestingly, the best performance on Cora is obtained when the coarsening rate is 1.00, with which Coarformer becomes equivalent to +Graphormer and still performs well. Meanwhile, the performance drops slightly as the coarsening rate decreases. However, even when the coarsening rate decreases to that corresponding to a coarse graph with size $n' \approx \sqrt{n}$, the performance of Coarformer still surpasses that of GCN/GAT. As for Chameleon, VN performs best when the corresponding $n'$ is close to $\sqrt{n}$. As for VE and Alg JC, they also show competitive results. Based on these results, we recommend attempting the coarsening rate corresponding to a $n'$ around $\sqrt{n}$ when applying Coarformer for large graphs.

Table 5: Sensitivity analysis: Mean accuracy (%) ± 95% confidence interval. Boldface letters are used to mark the best results.

| Datasets | Algorithm | $c = 0.01$ | | $c = 0.10$ | | $c = 0.50$ | | $c = 1.00$ | |
|---|---|---|---|---|---|---|---|---|---|
|  |  | GCN | GAT | GCN | GAT | GCN | GAT | GCN | GAT |
| Cora | VN | **88.14±0.65** | 87.44±0.62 | 87.77±0.87 | 87.64±0.57 | **87.82±0.70** | 87.87±0.50 | **89.26±0.75** | 88.46±0.62 |
|  | VE | 87.59±0.63 | 87.80±0.73 | 87.44±0.57 | 87.72±0.40 | 87.59±1.33 | **88.08±0.59** | 88.93±0.74 | 88.49±0.59 |
|  | Alg JC | 87.93±0.55 | **87.88±0.53** | **88.08±0.75** | **87.88±0.64** | 87.32±0.70 | 88.01±0.62 | 88.64±0.64 | **89.26±0.34** |
| Chameleon | VN | **66.96±1.05** | **65.95±1.74** | **68.18±1.44** | **67.35±1.24** | 65.80±1.04 | 65.78±1.64 | 65.80±1.23 | **65.80±1.27** |
|  | VE | 64.11±1.33 | 65.27±1.61 | 64.51±1.65 | 64.97±1.42 | 65.32±1.29 | 65.82±1.83 | 66.08±1.32 | 66.50±1.59 |
|  | Alg JC | 66.32±1.37 | 65.56±0.52 | 65.93±1.40 | 65.80±0.75 | **65.86±1.20** | **65.91±1.48** | **66.19±1.22** | 66.15±1.62 |

## 6 CONCLUSION

In this paper, we propose a novel two-view architecture Coarformer, which employs a GNN-based module and a Transformer-based module to encode nodes from the local and global view, respectively. The global view is constructed by graph coarsening, which implicitly regularizes the applied Transformer-based module and improves its efficiency. To enable the mutual enhancement between these two views, we further develop a cross-view propagation scheme that is consistent with mini-batch training. Extensive experiments conducted on large real-world graphs verify the advantages of Coarformer against existing Transformer-based methods in terms of both performance and resources (i.e., running time and GPU memory consumption). In summary, Coarformer extends the scope of usage of Transformer, paving the road towards a fully Transformer-based model for graph data.

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

# A APPENDIX

## A.1 STUDY ABOUT SAMPLE COMPLEXITY

In Sec. 5.1.1, we compare different methods with a so-called dense split of the node set (i.e., 60%:20%:20%). To assess the sample complexity of considered methods, we further compare several of them on Cora with a sparse split, i.e., 2.5%:2.5%:95%. Except for these ratios, other settings are kept the same as Sec. 5.1.1. The results are shown in Table 6. As expected, the performances of all the methods decrease when the size of the training set is reduced. With the sparse split, GraphSAGE and Coarformer exhibit a decrease of less than 20%, while the performance of the pure Transformer-based method Graphormer almost drops to only a half of that with the dense split. This phenomenon suggests a higher sample complexity of Transformer compared with GNN. We also present the performances of Graphormer with different hyper-parameter configurations in Table 7. Unlike the conventional cases where there are four or five heads and a hidden dimension of 128/256, a relatively small capacity is suitable for Graphormer on Cora.

Table 6: Performance comparisons on Cora with different splits: Mean accuracy (%) ± 95% confidence interval. Boldface letters are used to mark the best results.

| Split | GraphSAGE | Coarformer | Graphormer |
|---|---|---|---|
| Sparse | **73.72±0.89** | 72.47±1.40 | 39.67±5.66 |
| Dense | 87.42±0.78 | **88.69±0.82** | 77.41±4.30 |

Table 7: Details about HPO for Graphormer on Cora.

(a) With sparse split (2.5%:2.5%:95%)

| #head\hidden | 32 | 64 |
|---|---|---|
| 1 | **39.67±5.66** | 37.40±5.25 |
| 2 | 36.37±5.67 | 37.43±6.13 |

(b) With dense split (60%:20%:20%)

| #head\hidden | 32 | 64 |
|---|---|---|
| 1 | 77.14±4.80 | 74.75±4.30 |
| 2 | **77.41±4.30** | 71.53±5.89 |

## A.2 DATASETS DESCRIPTION

See Table 8 for a detailed description of the medium-sized dataset and Table 9 for the OGB extremely large dataset. These datasets come from different domains, and the nodes and edges from different datasets represent different meanings, including citation graph (Cora, CiteSeer, and PubMed), co-purchase graph (Computers and Photo), Wikipedia graph (Chameleon and Squirrel), actors collaborating graph (Actor), Website graph (Texas and Cornell), and extremely large datasets (ogbn-arxiv and ogbn-products) provided by OGB team.

Table 8: Dataset statistics

|  | Cora | CiteSeer | PubMed | Computers | Photo | Chameleon | Actor | Squirrel | Texas | Cornell |
|---|---|---|---|---|---|---|---|---|---|---|
| #Nodes | 2,708 | 3,327 | 19,717 | 13,752 | 7,650 | 2,277 | 7,600 | 5,201 | 183 | 183 |
| #Edges | 5,278 | 4,522 | 44,324 | 245,861 | 119,081 | 31,371 | 26,659 | 198,353 | 279 | 277 |
| #Features | 1,433 | 3,703 | 500 | 767 | 767 | 2,325 | 932 | 2,089 | 1,703 | 1,703 |
| #Classes | 7 | 6 | 5 | 10 | 8 | 5 | 5 | 5 | 5 | 5 |

## A.3 EXPERIMENTAL DETAILS

### A.3.1 DETAILS ABOUT PERFORMANCE COMPARISONS

In this section, we describe the details of the experiment with parameter settings. It is worth noting that, unlike the GPR-GNN setup, we report the accuracy of the test set when the model achieves the highest accuracy in the validation set. In contrast, GPR-GNN reports the accuracy of the test set when the loss function is lowest in the validation set. We use accuracy with a 95% confidence interval, equal to the micro-f1 score when it is a single-label classification, as an evaluation metric of effectiveness and the average training time as an evaluation metric of efficiency. In addition, we

Table 9: OGB dataset statistics.

| | #Nodes | #Train | #Validation | #Test | #Edges | #Features | #Classes |
|---|---|---|---|---|---|---|---|
| ogbn-arxiv | 169,343 | 90,941 | 29,799 | 48,603 | 1,166,243 | 128 | 40 |
| ogbn-products | 2,449,029 | 196,615 | 39,323 | 2,213,091 | 123,718,280 | 100 | 47 |

perform HPO for all models with weight decay in $\{0.0, 0.0005\}$, coarsening rate in $\{0.1, 0.01\}$, dropout rate in $\{0.0, 0.5\}$, and learning rate in $\{0.002, 0.01, 0.05\}$. The number of head is all set to 4. The sizes of the hidden layers are all set to 64. Also, we set the depth of the GNN model to be two so that it is encouraged to focus on the local information of each node and avoid over-smoothing.In particular, for APPNP and GPR-GNN, we use their optimal hyper-parameters as given in the paper(Chien et al., 2020). In addition, each model is trained 1,000 epochs with early stops if the accuracy on the validation set does not decrease within 200 epochs.

### A.3.2 DETAILS ABOUT COMPARISONS ON OGB

We perform HPO for all models with weight decay in $\{0.0, 0.0005\}$, dropout rate in $\{0.0, 0.5\}$, batch size in $\{256, 512\}$, and learning rate in $\{0.002, 0.003, 0.01\}$. And hidden dim is set to 256, the depth of GNN-based module are all set to 3, the coarsening rate is 0.01 on ogbn-arxiv and 0.001 on ogbn-products.

### A.3.3 DETAILS ABOUT SENSITIVITY ANALYSIS

We conduct experiments on a typical homophilic graph, Cora, and a typical heterophilic graph, Chameleon. For each coarsening algorithm, we specify different coarsening rates, where the coarsening rate, noted as $c$, takes value from $\{0.01, 0.1, 0.5, 1\}$. Meanwhile, to ensure the generality of our analysis, we consider both GCN and GAT as the GNN-based module in the fine-grained local view of Coarformer.

### A.4 EFFICIENCY COMPARISONS

These are the results of the efficiency of different local views of GNN-based module, the average training time per epoch (ms) and average total training time (s) are reported on Table 10 and Table 11.

Table 10: Efficiency comparisons on homophilic graphs: Average training time per epoch (ms)/total training time (s). Underlined letters indicate results under mini-batch training.

| | Cora | | CiteSeer | | PubMed | | Computers | | Photo | |
|---|---|---|---|---|---|---|---|---|---|---|
| | w/o | w/ | w/o | w/ | w/o | w/ | w/o | w/ | w/o | w/ |
| GCN | 3.81/0.83 | 10.26/2.28 | 4.32/0.95 | 7.28/1.54 | 4.46/0.94 | 10.87/2.33 | 5.3/1.25 | 8.62/2.06 | 4.44/1.14 | 7.64/1.87 |
| GIN | 3.43/0.7 | 6.68/1.35 | 4.72/0.95 | 8.25/1.67 | 5.47/1.15 | 10.41/2.21 | 17.63/5.16 | 22.45/5.04 | 9.61/3.42 | 12.36/2.99 |
| GAT | 6.36/1.36 | 9.92/2.04 | 6.2/1.31 | 10.29/2.11 | 6.8/1.41 | 18.88/4.06 | 7.08/1.81 | 10.2/2.3 | 6.39/1.56 | 10.33/2.4 |
| GT | 9.05/1.87 | 14.23/2.96 | 12/2.51 | 18.93/4.09 | 16.75/3.51 | 21.46/4.61 | 2512.29/520.78 | 2639.13/537.56 | 43.13/10.01 | 49.27/12.42 |
| APPNP | 5.69/1.22 | 8.81/2.06 | 5.81/1.27 | 8.71/2.32 | 5.88/1.26 | 9.34/2.17 | 6.18/1.8 | 9.38/3.2 | 6.06/1.7 | 8.93/3.03 |
| GPR-GNN | 6.4/1.31 | 9.34/1.94 | 6.54/1.34 | 9.72/2.05 | 6.87/1.47 | 9.42/2.18 | 7.56/1.88 | 10.44/2.48 | 6.56/1.45 | 9.61/2.28 |

Table 11: Efficiency comparisons on heterophilic graphs: Average training time per epoch (ms)/average total training time (s). Underlined letters indicate results under mini-batch training.

| | Chameleon | | Actor | | Squirrel | | Texas | | Cornell | |
|---|---|---|---|---|---|---|---|---|---|---|
| | w/o | w/ | w/o | w/ | w/o | w/ | w/o | w/ | w/o | w/ |
| GCN | 4.26/1.01 | 7.15/1.45 | 4.28/0.87 | 11.34/2.39 | 4.81/1.14 | 7.7/1.56 | 4.35/0.88 | 10.38/2.26 | 4.41/0.9 | 10.25/2.17 |
| GIN | 7.57/1.79 | 10.2/2.19 | 4.92/1.1 | 11.32/3.09 | 30.17/11.59 | 32.79/7.98 | 3.28/0.67 | 6.96/1.41 | 3.3/0.68 | 10.57/2.33 |
| GAT | 6.41/1.51 | 10.29/2.08 | 6.28/1.27 | 13.07/2.65 | 6.38/1.29 | 13.71/2.77 | 6.56/1.35 | 12.5/2.76 | 6.33/1.3 | 12.11/2.45 |
| GT | 31.46/6.36 | 35.64/7.2 | 16.29/3.3 | 22.08/4.59 | 669.13/159.40 | 689.16/164.63 | 8.86/1.82 | 10.53/2.17 | 8.54/1.76 | 10.46/2.27 |
| APPNP | 5.65/1.16 | 9.58/2.1 | 6.18/1.25 | 9.05/1.99 | 5.89/1.2 | 8.72/1.95 | 5.76/1.18 | 8.94/1.89 | 5.77/1.18 | 8.69/2.33 |
| GPR-GNN | 6.72/1.54 | 10.15/2.05 | 7.19/1.59 | 10.3/2.5 | 6.35/3.81 | 9.07/3 | 6.69/1.46 | 9.72/2.13 | 6.74/1.39 | 9.84/2.12 |

### A.5 SUPPLEMENTARY DATASETS EXPERIMENTAL RESULTS

In this section, we provide experimental results on the Computers, Photo, Texas and Cornell. All experimental settings are consistent with section 5.1.

### A.5.1 PERFORMANCE COMPARISONS BETWEEN DIFFERENT TRANSFORMER-BASED METHODS ON SUPPLEMENTARY DATASETS

Due to space limitations, we summarize the experimental results about performance comparisons between different Transformer-based methods for more datasets at Table 12.

Table 12: Performance comparisons between different Transformer-based methods: Mean accuracy (%) ± 95% confidence interval. Boldface letters are used to mark the best results.

| | | Computers | Photo | Texas | Cornell |
|---|---|---|---|---|---|
| Local | GraphSAGE | 89.65±0.26 | 93.84±0.40 | 89.18±2.58 | 85.08±4.64 |
| | GT | 85.28±0.22 | 93.77±0.46 | **93.77±1.63** | 88.03±4.04 |
| Global | Graphormer (w/o) | 80.95±2.29 | 82.26±1.08 | 37.70±29.85 | 54.75±31.03 |
| | Graphormer (PPR) | 81.98±2.07 | 78.05±7.51 | 37.70±29.85 | 53.77±33.21 |
| | Graphormer (SPD) | 80.36±1.05 | 80.61±4.17 | 38.36±29.26 | 55.08±31.27 |
| | Graphormer (All) | 80.92±1.58 | 83.38±2.12 | 37.70±29.85 | 55.08±31.27 |
| Local (GraphSAGE) + Global (Xformer) | +Graphormer (All) | 89.18±0.20 | 94.89±0.12 | 88.03±1.02 | 85.57±1.18 |
| | +Graph-Bert | 89.43±0.24 | **95.18±0.15** | 88.36±0.84 | 83.11±2.80 |
| | +Reformer | 88.59±0.52 | 94.59±0.48 | 86.56±2.64 | 83.11±4.29 |
| | +RT | 88.53±0.43 | 94.44±0.26 | 85.57±0.76 | 85.90±2.62 |
| | Coarformer (-CP) | 88.63±0.20 | 93.85±0.35 | 86.56±3.01 | 84.59±3.28 |
| | Coarformer | **89.78±0.26** | 94.69±0.39 | 89.18±2.04 | **88.52±2.23** |

### A.5.2 PERFORMANCE COMPARISONS ON DIFFERENT GNN-BASED MODULE ON SUPPLEMENTARY DATASETS

Due to space limitations, we summarize the experimental results about performance comparisons on different GNN-based module for more datasets at Table 13.

Table 13: Performance comparisons on different GNN-based module: Mean accuracy (%) ± 95% confidence interval. Boldface letters are used to mark the improvements.

| | | GCN | GIN | GAT | GT | APPNP | GPR-GNN |
|---|---|---|---|---|---|---|---|
| Computers | w/o | 83.75±0.47 | 58.16±1.00 | 87.21±0.17 | 85.28±0.22 | 86.38±0.39 | 88.70±0.45 |
| | w/ | 90.04±0.30 | 77.56±0.65 | 89.94±0.26 | 90.05±0.33 | 86.80±0.42 | 89.24±0.36 |
| | Δ | **6.29** | **19.40** | **2.73** | **4.77** | **0.42** | **0.54** |
| Photo | w/o | 90.52±0.42 | 39.76±1.17 | 91.9±0.73 | 93.77±0.46 | 93.38±0.40 | 93.90±0.43 |
| | w/ | 93.68±0.38 | 90.77±0.49 | 93.67±0.37 | 94.59±0.34 | 93.73±0.49 | 94.24±0.35 |
| | Δ | **3.16** | **51.01** | **1.77** | **0.73** | **0.35** | **0.34** |
| Texas | w/o | 73.28±2.45 | 80.00±1.56 | 79.18±2.49 | 93.77±1.63 | 88.36±2.59 | 91.48±2.02 |
| | w/ | 89.02±1.58 | 87.54±1.71 | 88.69±1.79 | 91.80±2.03 | 89.67±4.26 | 93.11±1.75 |
| | Δ | **15.74** | **7.54** | **9.51** | -1.97 | **1.31** | **1.63** |
| Cornell | w/o | 68.03±6.33 | 78.69±2.36 | 75.90±4.48 | 88.03±4.04 | 90.00±2.71 | 89.67±2.65 |
| | w/ | 87.38±3.43 | 88.20±2.07 | 87.54±2.73 | 90.49±2.02 | 89.69±2.42 | 90.00±2.93 |
| | Δ | **19.35** | **9.51** | **11.64** | **2.46** | -0.31 | **1.63** |

