# OpenReview forum: "Coarformer: Transformer for large graph via graph coarsening"
_ICLR.cc/2022/Conference — ICLR 2022 Submitted_

### Official Review · Reviewer_ZeUY · 2021-10-28

**Correctness:** 3
**Technical Novelty And Significance:** 3
**Empirical Novelty And Significance:** 3
**Recommendation:** 8
**Confidence:** 4

**Main Review:**

Strong points
* I find the empirical part of the paper solid. Training is performed over many splits. The model is tested on several datasets and many baselines and ablations are presented.
* The model is more memory efficient than other baselines which might be important in practical applications.

Weak points:
* Most of the results where Coarformer outperforms other baselines are not statistically significant (e.g. in Tab 1 and 3), there is usually some baseline that has mean value inside the confidence interval of Coarformer. However overall the model performs strongly and it is great that authors report the confidence intervals.
* The idea of using hierarchical graph representations to bridge long range dependencies isn't new, however I feel that prior work isn't properly discussed, other hierarchical graph models are missing in the Related Work section. Papers like DiffPool https://arxiv.org/abs/1806.08804 are highly relevant in my opinion. Other relevant papers are e.g. https://aclanthology.org/2020.emnlp-main.710.pdf or https://arxiv.org/pdf/2105.03388.pdf

Question:
* Sec 3.1 --- Is use of PPR as attention bias in graph transformers new? No citation is in that paragraph.
* Loss on coarse graph is just auxiliary, correct?
* How does the number of Coarformer's parameters compare to other compared models? Is lower memory consumption just a function of fewer parameters or is it solely due to the different structure of the computation graph?
* Why are you testing just transformer based architectures on the global scale? E.g. GraphSAGE can be used for both local and global level.


Presentation improvements:

* BERT (Kenton & Toutanova, 2019) is a strange citation for BERT, I was expecting Devlin et al.
* Cite PPR.
* Sec 3.1 says that \Beta is a learnable param, however Sec 3.3 says it is set to 1 for simplicity. I would omit \Beta completely since I don't see any value in it. In every formula one can add similar hparams that are set to 1 and they don't add any value to the model.
* Related Work (no S)
* Define HPO, I get that it is hyper param optimization but it isn't explicitly stated.
* Cite SPD.
* Add at least one sentence describing VN, VE and Alg JC and cite them.
* Instead of Tab 5 a graph with 'c' on the x axis would be more appropriate. It will be easier to interpret the data and see possible trends in it.
* Table 12, Cornell column: Coarformer is in bold even though GT has better result.

**Summary Of The Paper:**

The paper proposes hierarchical neural network model for processing graphs. The lower level can use any GNN model suitable for processing smaller subgaphs of the original graph while the higher level part realized by a transformer operates on an abstracted 'coarse' graph.

**Summary Of The Review:**

I am slightly leaning towards acceptance. While I think that related work should be better discussed I think that many people can be interested in low memory overhead of the model and the whole community can benefit from that.

--------------- Post Rebuttal -------------------
After the latest round of improvements I believe the paper is above the acceptance bar.

---

> ### Author Response · Authors · 2021-11-22
> **Response to Reviewer ZeUY**
>
> Many thanks for your valuable feedback and comments! Thanks for pointing out the grammatical errors and citation errors, and we have revised the paper based on your comments.
>
> **W1**: We would like to clarify that the improvement of our Coarformer over SOTA models is comparable to the improvement of the SOTA model [1] over their baselines (e.g., on Cora, GPR-GNN [1] and Coarformer outperforms their strongest baseline by 0.10 and 0.45, respectively; on Computers, GPR-GNN and Coarformer outperforms their strongest baseline by -0.21 and 0.54, respectively; on Actor, GPR-GNN and Coarformer outperforms their strongest baseline by 0.44 and 0.93, respectively). In addition, we optimize the implementation of Transformer for some data sets, add a linear layer before Transformer. And we performed a hyperparameter optimization under the same settings. The updated results are shown in Table 1 below, which have been revised in Table 3 and Table 13 in our paper. As can be seen from the table, Coarformer statistically significantly outperforms other SOTA models.
>
> **Table 1 Improvement of Mean Classification Accuracy (Percent)**
>
> |           | GCN-$\Delta$ | GIN-$\Delta$ | GAT-$\Delta$ | GT-$\Delta$ | APPNP-$\Delta$ | GPR-GNN-$\Delta$ |
> | --------- | ---------- | ---------- | ---------- | --------- | ------------ | -------------- |
> | Cora      | 0.56       | 1.52       | 0.65       | -0.43     | 0.68         | 0.45           |
> | CiteSeer  | -1.83      | 4.04       | -1.65      | -1.53     | -0.33        | -0.43          |
> | PubMed    | 2.90       | -0.95      | 1.84       | 1.33      | 0.37         | 0.36           |
> | Computers | 6.29       | 19.40      | 2.73       | 4.77      | 0.42         | 0.54           |
> | Photo     | 3.16       | 51.01      | 1.77       | 0.73      | 0.35         | 0.34           |
> | Chameleon | 7.87       | 25.01      | 7.96       | 6.71      | 6.09         | 0.11           |
> | Actor     | 3.57       | 0.05       | -0.77      | 0.73      | 0.66         | 0.93           |
> | Squirrel  | 7.61       | 23.81      | 3.71       | -2.08     | 10.40        | -2.69          |
> | Texas     | 15.74      | 7.54       | 9.51       | -1.97     | 1.31         | 1.63           |
> | Cornell   | 19.35      | 9.51       | 11.64      | 2.46      | -0.31        | 1.63           |
>
> **W2**: We agree that the papers you mentioned should be better discussed in the related work, and we have revised the paper based on your comments. Here we would like to clarify the differences between our work and these works and restate the novelty of our two-view architecture Coarformer. DiffPool [2] hierarchically pools the graph. HGN [3] connects the node (word) to the supernode (paragraphs or topics). HGNN [4] builds connections between each layer of GNN. In contrast, our work aims to generalize Transformer to work on large graphs. No existing Transformer-based work can achieve competitive performance on large-scale graphs, such as OGBN-Products with millions of nodes, due to: (1) The vast number of distant nodes distract the necessary attention of each target node from its local neighborhood; (2) The quadratic computational complexity regarding the number of nodes makes the learning procedure costly. Therefore, in this paper, we introduce GNNs that can efficiently encode local information to solve the first problem for the Transformer. As for the issue of computational complexity, we introduce coarsening algorithms to solve it. This down-sampling method preserves coarse global information for Transformer. In summary, we present a novel two-view architecture Coarformer with a novel CROSS-VIEW PROPAGATION scheme to mutually enhance each view.

---

> > ### Author Response · Authors · 2021-11-22
> > **Response to Reviewer ZeUY (2)**
> >
> > **Q1**: Yes, it is new. We use different Spatial Encoding for Graphormer, and finally, we chose PPR instead of SPD, partly because the time to compute SPD is too large. And PPR has many approximation algorithms, which cost much less.
> >
> > **Q2**: Yes, the loss on the raw graph plays a dominant role.
> >
> > **Q3**: Both. Coarformer uses fewer parameters, and the size of the required GPU memory of the computation graph is smaller. The number of parameters for each model is shown in Table 2 below.
> >
> > **Table 2 Number of parameters**
> >
> > |               | Coarformer | +Graphormer | +Reformer | +Routing Transformer |
> > | ------------- | ---------- | ----------- | --------- | -------------------- |
> > | #Num of param | 191K       | 192K        | 192K      | 192K                 |
> >
> > **Q4**: The sparsity of the graph changes after coarsening, i.e., the coarse graph has a denser adjacency matrix, and connected components with few nodes become isolated nodes after coarsening. In other words, GNNs lose the ability to model graph structure on almost fully connected graphs or those isolated nodes without neighbors in coarse graphs. Therefore, considering this phenomenon in the real world, we prefer to use the Transformer-base model on the coarse graph, which can uncover the potential connections between nodes better than the GNN-based model.
> >
> > F0-1&3&7-8: Thanks to the improvements you have suggested, we have revised our paper and cited the following papers: BERT [5], PPR [6].
> >
> > **F2**: \Beta in Section 3.3 should be changed to \gamma, and when $\gamma=1$, the loss on the raw graph plays a dominant role.
> >
> > **F4&5**: HPO stands for Hyper Param Optimization, and SPD stands for Shortest Path Distances. We have revised our paper following the suggestions.
> >
> > **F6**: The description and citation can be seen in Appendix A.3.3, which we have moved to the body text in the final version.
> >
> > [1] Eli Chien, et al. Adaptive universal generalized PageRank graph neural network. In International Conference on Learning Representations, 2020.
> >
> > [2] Ying, Rex, et al. Hierarchical graph representation learning with differentiable pooling. In NeurIPS. 2018.
> >
> > [3] Fang, Yuwei, et al. Hierarchical Graph Network for Multi-hop Question Answering. In EMNLP, 2020.
> >
> > [4] Stanislav Sobolevsky. Hierarchical graph neural networks.CoRR, abs/2105.03388, 2021.
> >
> > [5] Devlin, Jacob, et al. BERT: Pre-training of Deep Bidirectional Transformers for Language Understanding. In NAACL-HLT, 2019.
> >
> > [6] Lawrence Page, Sergey Brin, Rajeev Motwani, and Terry Winograd. The pagerank citation ranking: bringing order to the web. 1999.

---

> > > ### Comment · Reviewer_ZeUY · 2021-11-25
> > > **Response**
> > >
> > > Thank you, I think that the paper is now better positioned in relation to related work and the other improvements also made it stronger.

---

### Official Review · Reviewer_1q2q · 2021-11-01

**Correctness:** 3
**Technical Novelty And Significance:** 2
**Empirical Novelty And Significance:** 2
**Recommendation:** 5
**Confidence:** 3

**Main Review:**

Overall, this paper tackles an important question on scaling Transformer to large graphs.
The proposed method is technically sound.
The experimental results are not very strong (performance gain over baselines are marginal).

With that being said, I have the following concerns for the paper:

1 The proposed Coarformer is not well motivated and seems overly complicated.
It seems that the main contribution is to operate the Transformer on a coarsened graph.
The rest of the method is quite standard.
I think a simpler model to discuss is to use the GNN node embeddings as the input to the coarsened graph, or directly combine the predictions from both models.

2 Proposition 1 is not well explained. I'm not sure why it can lead to the claim that "Coarformer can cooperate with mini-batch training and thus is viable on large graphs".

3 How important is the graph coarsening algorithm? How scalable is the algorithm being used?



**Summary Of The Paper:**

This paper proposes Coarformer, a two-view architecture that consists of a standard GNN learning pipeline, and a Transformer over a coarsened graph.

This paper represents an interesting idea that scales Transformer to large graphs.

**Summary Of The Review:**

Overall, I like the motivation of the paper. The proposed methodology should be improved though. And the experimental results are not very strong.

---

> ### Author Response · Authors · 2021-11-22
> **Response to Reviewer 1q2q**
>
> Many thanks for your valuable feedback and comments! However, we would like to clarify that the accuracy of the improvement of our Coarformer over SOTA models is comparable to the improvement of the SOTA model [1] over their baselines  (e.g., on Cora, GPR-GNN [1] and Coarformer outperforms their strongest baseline by 0.10 and 0.45, respectively; on Computers, GPR-GNN and Coarformer outperforms their strongest baseline by -0.21 and 0.54, respectively; on Actor, GPR-GNN and Coarformer outperforms their strongest baseline by 0.44 and 0.93, respectively).
> Further, to address your concern about the accuracy of our two-view architecture, we optimize the implementation of Transformer for some datasets, add a linear layer before Transformer. And we perform hyperparameter optimization under the same settings. The updated results are shown in Table 1 below, which have been updated in Table 3 and Table 13 in our paper. As can be seen from the table, Coarformer statistically significantly outperforms other SOTA models. If our clarifications have addressed your concerns, could you please kindly consider increasing the overall score?
>
> **Table 1 Improvement of Mean Classification Accuracy (Percent)**
>
> |           | GCN-$\Delta$ | GIN-$\Delta$ | GAT-$\Delta$ | GT-$\Delta$ | APPNP-$\Delta$ | GPR-GNN-$\Delta$ |
> | --------- | ---------- | ---------- | ---------- | --------- | ------------ | -------------- |
> | Cora      | 0.56       | 1.52       | 0.65       | -0.43     | 0.68         | 0.45           |
> | CiteSeer  | -1.83      | 4.04       | -1.65      | -1.53     | -0.33        | -0.43          |
> | PubMed    | 2.90       | -0.95      | 1.84       | 1.33      | 0.37         | 0.36           |
> | Computers | 6.29       | 19.40      | 2.73       | 4.77      | 0.42         | 0.54           |
> | Photo     | 3.16       | 51.01      | 1.77       | 0.73      | 0.35         | 0.34           |
> | Chameleon | 7.87       | 25.01      | 7.96       | 6.71      | 6.09         | 0.11           |
> | Actor     | 3.57       | 0.05       | -0.77      | 0.73      | 0.66         | 0.93           |
> | Squirrel  | 7.61       | 23.81      | 3.71       | -2.08     | 10.40        | -2.69          |
> | Texas     | 15.74      | 7.54       | 9.51       | -1.97     | 1.31         | 1.63           |
> | Cornell   | 19.35      | 9.51       | 11.64      | 2.46      | -0.31        | 1.63           |
>
> **F1**: Thanks for raising the concern. We have conducted the ablation study of Coarformer compared to the simple mixture (i.e., “Coarformer (-CP)”) that has no interaction between the two views. The results are shown in Table 1 in our paper, where the simple mixture outperforms than GNNs in terms of accuracy, but underperforms Coarformer. In addition, the Transformer-based module of Coarformer works on a coarse graph to capture global information, which significantly reduces the complexity of time and space compared to the simple mixture that trivially applies Transformer to the original graph.
>
> **F2**: As described in “Sampling-based cross-view propagation on large graphs”, full-batch training is infeasible on large graphs. In this section, we prove that the GraphSAINT sampler we use for GNN in local view is still an unbiased estimator in the coarse global view. Thus, Coarformer can be trained with both the full-batch setting and the mini-batch setting.
>
> **F3**: In section 5.3, we conducted experiments on the sensitivity of Coarformer to coarsening algorithms. We evaluate Coarformer with different coarsening algorithms and different coarsening rates, and the results are shown in Table 5 of our paper. The experimental results illustrate that there is no apparent difference between different coarsening algorithms, indicating the robustness of Coarformer w.r.t. them. As for scalability, the memory consumption of the coarsening algorithm we use grows nearly linearly with the number of the edge [2].
>
> [1] Eli Chien, et al. Adaptive universal generalized PageRank graph neural network. In International Conference on Learning Representations, 2020.
>
> [2] Andreas Loukas. Graph reduction with spectral and cut guarantees. Journal of Machine Learning Research, 2019.

---

### Official Review · Reviewer_QYtX · 2021-11-01

**Correctness:** 3
**Technical Novelty And Significance:** 3
**Empirical Novelty And Significance:** 3
**Recommendation:** 6
**Confidence:** 5

**Main Review:**

### Strengths
* Application of transformer on large-graphs with graph coarsening is a relatively undiscovered problem (though not entirely novel, see below weaknesses).
* Information propagation across GNNs and transformer to capture both local and global information on graphs is novel and valuable.
* The paper is well-written and easy to understand.

### Weaknesses
* Using transformer on the compressed graph is already discovered idea [1]. Specifically, [1] captures the local information on the original graph with GNNs, and then further captures the global information on the compressed graph with transformer. Also, the idea of alleviating computational complexities, both memory and time, in the paper is similar to the mentioned previous work [1].
* In Table 2, it seems the authors do not incorporate the time for graph coarsening. If the authors compress the large graph, then the resulting computational cost for coarsening is likely to be large. What relative cost does the proposed method needs for graph coarsening?
* In Figure 2, the authors only display the GPU memory usage of small size graphs, namely Cora and Chameleon. I think the extra memory usage required for the proposed method is relatively increasing, when we use the large-size graphs that authors tackle. Thus, since the authors target the large graphs, I recommend the authors to change the datasets in Figure 2 to larger ones, such as OGB or PubMed.
* In Table 4, I think the experimental comparisons against baselines are not enough or entirely fair. The authors exclude the powerful transformer-based baseline, namely Graphormer, but also it seems the authors use a small number of batches for baselines (Correct me if I misunderstand).

### Typos
* In Section 3.3, I think the $\beta$ in the sentence "we adopt $\beta=1$ in our experiment" should be changed to the $\gamma$. The $\beta$ is already defined as the learnable parameter for the PPR matrix in Section 3.1.
* In the caption of Table 1, "." is missing at the end.

---
[1] Baek et al., Accurate Learning of Graph Representations with Graph Multiset Pooling, ICLR 2021.

---
**-------------------- After rebuttal --------------------**

Summing up all my reviews and follow-up comments, I think this paper is above the borderline, and I lean towards an acceptance of this work.

**Summary Of The Paper:**

This paper proposes to use the transformer architecture on the coarse graph obtained from the graph coarsening algorithm.

To be more specific,
* Since directly using the transformer architecture to the large-scale graph is computationally prohibitive, the authors use the existing graph coarsening algorithms to the large graphs, and then use the transformer on the coarse graphs, which can capture the global information of the given graph in contrast to the GNNs capturing local information.
* To simultaneously use both local and global structures of the graph, the authors first separately capture the local and global information of the given graph with GNNs on the original graph and transformers on the coarse graph, respectively, and then propagate information of local and global to each other with the cross-view propagation scheme.
* The authors show that the proposed Coarformer outperforms baselines on node classification tasks, and also show that the proposed model is highly efficient against the global transformer models for graphs.

**Summary Of The Review:**

I think this is a borderline paper, as graph coarsening to use the transformer on large graphs is not entirely novel, however, it also has clear merits: a novel scheme of information propagation across original and coarse graphs; an almost complete piece of work except for some minor issues in the weaknesses above.

---

> ### Author Response · Authors · 2021-11-22
> **Response to Reviewer QYtX**
>
> Many thanks for the valuable feedback and insightful comments! We are glad that you found our work well-written and our CROSS-VIEW PROPAGATION scheme novel. We have revised our paper following the suggestions and addressed all of your comments in the following response:
>
> **W1**: Thank you for the comment! We agree that using Transformer on the compressed graph has already been discussed to capture global information, but we would like to clarify that our work aims to generalize Transformer to work in large graphs. No existing Transformer-based work can achieve competitive performance on large-scale graphs, such as OGBN-Products with millions of nodes. The reason is two-fold: (1) The vast number of distant nodes distract the necessary attention of each target node from its local neighborhood; (2) The quadratic computational complexity regarding the number of nodes makes the learning procedure costly. Therefore, in this paper, we introduce GNNs that can efficiently encode local information to solve the first problem for the Transformer. As for the issue of computational complexity, we introduce coarsening algorithms to solve it. This down-sampling method preserves coarse global information for Transformer. [1] proposes an attention-based pooling algorithm GMT. GMT encodes node features using GNN, then reconstructs node relationships using the attention mechanism to achieve pooling purposes. However, our Coarformer is not a dedicated pooling algorithm for graph classification but a general architecture combining local and global information with great scalability and competitive practical performance. Finally, thanks for your reminder, and we have discussed the difference between our work and the works you mentioned in the latest version of our submission.
>
> To summarize our novelties:
>
> Scalability aspect: No existing Transformer-based work can achieve competitive performance on large-scale graphs, such as OGBN-Products with millions of nodes. They apply Transformer to either molecular graphs with just tens of nodes or medium-sized graphs (e.g., Coar) with local receptive fields. In contrast, we empirically show the superiority of Coarformer on large graphs.
>
> Architecture aspect: No prior work models local view and global view with GNN and Transformer, respectively. We notice the complementary of these two kinds of neural architectures and design Coarformer for allowing them to play to their strength. Moreover, no prior work has exploited the benefit of interactions between the local and global views. Instead, our proposed CP achieves the mutual enhancement between these two views, contributing to Coarformer's superior performance.
>
> **W2**: Thanks for raising the concern. We appreciate you mentioning [1] in W1, and we have taken learnable coarsening into account in developing Coarformer. However, this would incur significant time consumption on large graphs. Therefore, to reduce the complexity of the Transformer and thus capture the global information efficiently, we use a graph coarsening algorithm [2]. The graph coarsening is performed only on the CPU and in the pre-processing stage before our training course. The additional time consumption of coarsening can be found in Figure 3 of [2].
>
> **W3**: We are glad that you pointed this out! We should have used a larger dataset to demonstrate the GPU memory advantage that our Coarformer brings. However, due to hardware limitations (maximum GPU memory is 11G), other Transformer-based models would fail because of OOM with the full-batch training in PubMed. Thus, we decide to use Cora and Chameleon to demonstrate the benefits of our consumption advantage under the same settings.
>
> **W4**: Yes, because the GPU memory required for full-batch training Graphormer is too large, we have to consider a mini-batch setting for Graphormer on some graphs. This mini-batch setting might be the reason for the inferior result of Graphormer on these graphs. However, due to time constraints, we will only be able to update this result by using a GPU with more memory for the experiments in the future.
>
> If our clarifications have addressed your concerns, could you please kindly consider increasing the overall score?
>
> [1] Baek et al., Accurate Learning of Graph Representations with Graph Multiset Pooling, ICLR 2021.
>
> [2] Andreas Loukas. Graph reduction with spectral and cut guarantees. Journal of Machine Learning Research, 2019.

---

> > ### Comment · Reviewer_QYtX · 2021-11-22
> > **Thank you for your response**
> >
> > I really appreciate your comments. Generally speaking, I'm satisfied with the explanations, and the details are below:
> >
> > **QW1.** I agree that the generalization of Transformer to large graphs is a relatively undiscovered but highly important problem. However, the motivations: the vast number of nodes distract the necessary attention, and the quadratic computational complexity, are already developed concepts [1]. However, in contrast to GMT [1] that is dedicated to graph pooling, the proposed algorithm is a general method that could be used for node and graph-level tasks, thanks to the cross-view propagation scheme. I acknowledge the novelty of it, and also appreciate the comparison against the GMT [1] in the revision.
> >
> > **QW2.** Thank you for your explanation on graph coarsening, and its time cost. However, I believe it is better to incorporate the coarsening time in the paper, rather than only citing the related work. This makes me believe that coarsening time is much larger than the running time in Table 2.
> >
> > **QW3.** Thank you for your explanation. In this case, I think the used datasets, namely Cora and Chameleon, are enough for my score judgment, while the other datasets are highly appreciated if possible though.
> >
> > **QW4.** Thank you for your explanation. Please carefully describe the setting you used as the important hyper-parameters (e.g., batch size) are different across experiments. Also, if the batch size is different, then I recommend the authors set the batch size as the same for all experiments. Finally, the Graphormer is already not in Table 4, thus what do you refer to in your previous response?
> >
> > ---
> >
> > I think the novelty of this work is still mild, whereas, it seems it is over the acceptance bar. However, due to the remaining concerns in QW2 and QW4, I think this paper is not yet complete, and lies somewhere between 5 (borderline reject) and 6 (borderline accept). Thus, I have changed my score to 6 thanks to the authors' effort on clarifying weaknesses and questions, meanwhile, I'm not so confident on the acceptance.
> >
> > ---
> >
> > [1] Baek et al., Accurate Learning of Graph Representations with Graph Multiset Pooling, ICLR 2021.

---

> > > ### Author Response · Authors · 2021-11-23
> > > **Response to Reviewer QYtX (2)**
> > >
> > > Thanks again for your valuable feedback and insightful comments! For your remaining concerns in QW2 and QW4, we provide more detailed clarifications below:
> > >
> > > **QW2**: Thanks for your suggestion. Due to the limited time, we present the time consumption of executing the graph coarsening algorithm (Variation Neighborhoods [1]) on Cora and Chameleon in Table 1 below and the second paragraph of Section 5.1.1 "Results and analysis". We have revised "total running time" to "total training time" in our paper, which means the total training time includes the time to perform the training course, excluding the time for data pre-processing and inference. We agree that it is necessary to present the coarsening time so that the readers can clearly understand the time consumed by pre-processing and training, respectively. Meanwhile, we also want to emphasize that many steps in the machine learning pipeline, such as HPO, will be performed after data pre-processing with hundreds of training courses, but each training course does not require additional data pre-processing. As with the same coarsening algorithm and coarsening rate, the graph coarsening is only executed once and cached.
> > >
> > >
> > >
> > > **Table 1 Time consumption of data pre-processing (s).**
> > >
> > > | Dataset/Coarsening rate | 0.1  | 0.01 |
> > > | ----------------------- | ---- | ---- |
> > > | Cora                    | 4.01 | 4.04 |
> > > | Chameleon               | 5.02 | 5.04 |
> > >
> > >
> > >
> > > **QW4**: Thanks for raising the concern. We want to clarify that we tried to perform hyperparameter optimization for all models with batch size in {256, 512, 1024} in Table 4 in our paper, but due to hardware limitations (maximum GPU memory is 11G), only Coarformer can be conducted when batch size is 1024. Thus, to keep the comparison fair, we change the HPO space of batch size from {256, 512, 1024} to {256, 512} for all models including our Coarformer. And we think our experimental setup is fair where all models have the same HPO space. Moreover, the computation of SPD in Graphormer via Floyd-Warshall algorithm takes too long in OGBN-Products with millions of nodes, so we believe that Graphormer is not efficient enough for large graphs.
> > >
> > >
> > >
> > > [1] Andreas Loukas. Graph reduction with spectral and cut guarantees. Journal of Machine Learning Research, 2019.

---

> > > > ### Comment · Reviewer_QYtX · 2021-11-24
> > > > **Thank you for your response**
> > > >
> > > > Thank you for addressing my remaining concerns.
> > > >
> > > > **QW2:** I can now clearly see the total running time of the proposed algorithm. As shown in Table 1 of the previous response, the graph coarsening algorithm is indeed not efficient, i.e., the coarsening time of the given graph is much larger than the total training time of it. However, as this paper does not focus on developing novel coarsening algorithms, the slower speed of graph coarsening does not affect my score judgment. I rather appreciate that the authors make use of graph coarsening to perform Transformer on the compressed graph. Nevertheless, I suggest the authors to carefully incorporate the data pre-processing time (maybe for all datasets), and also discuss it, for example, if there is a more efficient coarsening algorithm, then the time for coarsening could be significantly reduced, in the next revision.
> > > >
> > > > **QW4:** Thank you for your response. I think now the experiment in Table 4 is fair across all models.
> > > >
> > > > ---
> > > >
> > > > The authors carefully address my concerns about QW2 and QW4 via the previous response, and I have no concerns for them. Summing up all my reviews and comments, I think this paper is above the borderline, and I lean towards an acceptance of this work.

---

> > > > > ### Author Response · Authors · 2021-11-25
> > > > > **Thanks for your valuable feedback!**
> > > > >
> > > > > Thanks again for your valuable feedback! We will update our paper following the suggestions in the next revision.

---

### Official Review · Reviewer_ErXY · 2021-11-02

**Correctness:** 3
**Technical Novelty And Significance:** 2
**Empirical Novelty And Significance:** 2
**Recommendation:** 3
**Confidence:** 4

**Main Review:**

strengths

1. The paper is easy to follow, the ideas combine the GNNs and Transformer is interesting, take benefits of the GNN, local aggregation and Transformer, global aggregation.

2. The coarsening part is good, to reduce the complexity of the self-attention in the Transformer.

3. A sampling techniques are also adopted for mini-batch training.


Weakness:
1. The novelty of this work is not very high, it combines a lot of existing of techniques, for example GNN, Transformer, Graph Coarsening, Sampling, PPR etc. These methods are widely explored[1, 2].  The overall design is very straightforward. Somehow I feel the model is a little heavy with more model parameters.

[1] https://openreview.net/forum?id=uxpzitPEooJ
[2] https://arxiv.org/pdf/2106.05150.pdf

2. The idea of the global and long range dependecies is also widely expolited, For example
[1] Geom-GCN: Geometric Graph Convolutional Networks
[2] Improving Breadth-Wise Backpropagation in Graph Neural Networks Helps Learning Long-Range Dependencies


**Summary Of The Paper:**

In this work, the authors mainly combine the GNN and Transformer to learn both local and global information in large-scale graph.

**Summary Of The Review:**

In summary, I think the idea of combine GNN and Transformer is very interesting. However, the novelty of this work is very straightforward, a mixture of many existing work.

---

> ### Author Response · Authors · 2021-11-22
> **Response to Reviewer ErXY**
>
> Many thanks for your valuable feedback and comments! We have revised our paper following the suggestions and addressed all of your comments in the following response:
>
> **W1**: Thanks for raising the concern. However, we would like to clarify that our work aims to generalize Transformer to work on large graphs. No existing Transformer-based work can achieve competitive performance on large graphs, such as OGBN-Products with millions of nodes, due to: (1) The vast number of distant nodes distract the necessary attention of each target node from its local neighborhood; (2) The quadratic computational complexity regarding the number of nodes makes the learning procedure costly. Therefore, in this paper, we introduce GNNs that can efficiently encode local information to resolve the first issue for the Transformer. As for the issue of computational complexity, we adopt coarsening algorithms to preserve coarse global information while reducing the number of tokens for Transformer. Further, we present a novel two-view architecture Coarformer with a novel CROSS-VIEW PROPAGATION (CP) scheme to mutually enhance each view, instead of a simple mixture of GNN and Transformer.
> Although each individual component might have been explored by some existing papers, the organic overall design makes them work together to tackle the challenges when generalizing Transformer to large graphs, which is the main contribution of our work. From the perspective of scalability, existing works in the line of Graph Transformer apply Transformer to either molecular graphs with just tens of nodes or medium-sized graphs (e.g., Cora) with local receptive fields. In contrast, we empirically show the superiority of Coarformer on large graphs. Such progress is novel to the community.
>
> We agree that Coarformer brings more parameters than GNNs. However, compared to the normal GNN-based model, there is only a slight increase in the computation time and GPU memory usage of our Coarformer. And Figure 2 and Table 2 of our paper illustrate that Coarformer does not bring too much extra time and space consumption.
>
> **W2**: We agree that some works have attempted to obtain the long-range dependencies from graphs, such as Geom-GCN and ResRGAT. Geom-GCN maps the graph to a latent space, and ResRGAT relies on attention-based aggregation to reduce exponential gradient decay with breadth-wise residual connections. However, our two-view architecture Coarformer is orthogonal to these GNNs. We use Transformer - self-attention mechanism on a fully connected computation graph with a coarsening algorithm to obtain the long-range information. Considering your suggestion, we report Geom-GCN's best performance cited from [1] [2] and present our results of Coarformer with the same experimental setup as theirs in Table 1 below. The experiments demonstrate that Coarformer is much more competitive than Geom-GCN and the state-of-the-art model GPR-GNN.
>
> **Table 1 Mean Classification Accuracy (Percent)**
>
> |                      | Cora  | CiteSeer | PubMed | Computers | Photo | Chameleon | Actor | Squirrel | Texas | Cornell | Wisconsin |
> | -------------------- | ----- | -------- | ------ | --------- | ----- | --------- | ----- | -------- | ----- | ------- | --------- |
> | Geom-GCN             | 85.27 | 77.99    | 90.05  | OOM       | OOM   | 60.90     | 38.14 | 31.63    | 67.57 | 60.81   | 64.12     |
> | GPR-GNN              | 88.48 | 78.49    | 90.90  | 88.70     | 93.90 | 66.63     | 40.74 | 52.31    | 91.48 | 89.67   | 90.25     |
> | Coarformer (GPR-GNN) | 88.93 | 78.06    | 91.26  | 89.24     | 94.24 | 66.74     | 41.67 | 51.85    | 93.11 | 90.00   | 91.38     |
>
> If our clarifications have addressed your concerns, could you please kindly consider increasing the overall score?
>
> [1] Pei, Hongbin, et al. Geom-GCN: Geometric Graph Convolutional Networks. In International Conference on Learning Representations, 2019.
>
> [2] Eli Chien, et al. Adaptive universal generalized PageRank graph neural network. In International Conference on Learning Representations, 2020.

---

### Public Comment · ~Benedek_Andras_Rozemberczki1 · 2021-11-14
**Squirrel and Chameleon Datasets**

The datasets in the title were introduced in this paper:

Please add this citation:

@article{rozemberczki2021multi,
  title={Multi-scale attributed node embedding},
  author={Rozemberczki, Benedek and Allen, Carl and Sarkar, Rik},
  journal={Journal of Complex Networks},
  volume={9},
  number={2},
  pages={cnab014},
  year={2021},
  publisher={Oxford University Press}
}

---

> ### Author Response · Authors · 2021-11-22
> **Update Citation**
>
> Thanks, we have added this citation in the latest version of the paper.

---

### Decision · Program_Chairs · 2022-01-20

**Decision:**

Reject

**Comment:**

The paper aims to scale transformers to large graphs. In this regard, authors propose to first obtain a "coarse" version of the large graph using existing algorithms. With reduced number of nodes in the coarse graph, we can employ the transformer efficiently to capture the global information. To capture the local information, GNNs are employed. Finally, authors carry out extensive experiments on a range of graph datasets. Also, reviewers do appreciate reporting the confidence intervals. We thank the reviewers and authors for engaging in an active discussion. Unfortunately, the reviewers are in a consensus that novelty of the proposed method is limited: it is combination of existing techniques and similar ideas have been widely used in the literature. Also, the empirical results are not very significant. Thus, unfortunately I cannot recommend an acceptance of the paper in its current form.